# DIFFUDETR: RETHINKING DETECTION TRANSFORMERS WITH DENOISING DIFFUSION PROCESS

**Youssef Nawar**[* 1,2,3] **Mohamed Badran**[* 1,3] **Marwan Torki**[1]
[1] Alexandria University, [2] Technical University of Munich , [3] Applied Innovation Center
`youssof.nawar@tum.de`,{`es-mohamedmostafabadran, m.torki`}`@alexu.edu.eg`

## ABSTRACT

In this paper, we present DiffuDETR, a novel approach that formulates object detection as a conditional object query generation task, conditioned on the image and a set of noisy reference points. We integrate DETR-based models with denoising diffusion training to generate object queries' reference points from a prior gaussian distribution. We propose two variants: DiffuDETR, built on top of the Deformable DETR decoder, and DiffuDINO, based on DINO's decoder with contrastive denoising queries (CDNs). To improve inference efficiency, we further introduce a lightweight sampling scheme that requires only multiple forward passes through the decoder. Our method demonstrates consistent improvements across multiple backbones and datasets, including COCO 2017, LVIS, and V3Det, surpassing the performance of their respective baselines, with notable gains in complex and crowded scenes. Using ResNet-50 backbone we observe a **+1.0** in COCO-val, reaching 51.9 mAP on DiffuDINO compared to 50.9 mAP of the DINO. We also observe similar improvements on LVIS and V3DET datasets with **+2.4** and **+2.2** respectively. The project page and source code are available at `https://mbadran2000.github.io/DiffuDETR`.

## 1 INTRODUCTION

Object detection is a fundamental task in computer vision. It has gained much attention in recent years for its wide use in real-world applications. Object detection can be decomposed to two more primitive tasks: object localization and object classification. Traditional methods depend heavily on predefined bounding boxes (Liu et al., 2016), CPU-intensive selective search (Girshick et al., 2014), and region proposals networks (Girshick, 2015; Cai & Vasconcelos, 2018) to propose candidate locations. These methods, while effective, limit flexibility and generalizability in network training.

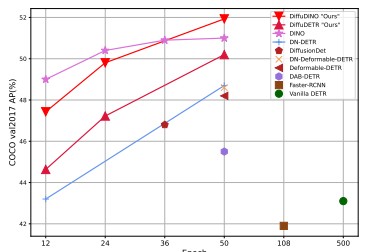

Figure 1: Performance of DiffuDINO against other DETR and CNN-based models using ResNet-50 (He et al., 2016) backbone.

Initial deep learning approaches (Girshick, 2015; Cai & Vasconcelos, 2018) require a set of predefined anchor boxes and heuristics to generate proposals. This method involves a one-to-many label assignment strategy, wherein each ground-truth bounding box is matched with multiple points in the detector's predictions. Despite their strong performance, these detectors depend heavily on several manually designed components, such as the predefined anchor boxes or non-maximum suppression for postprocessing.

Later, DEtection TRansformer (DETR) (Carion et al., 2020) proposed an end-to-end training objective by solving the problem as a bipartite set matching between a set of predictions and ground truths. These architectures rely on object queries, where each query matches exactly one object what is also known as one-to-one matching. This approach simplified the training objective and achieved SOTA results in both object detection. However, it suffers from query initialization problem.

---

[*]These authors contributed equally to this work.

Denoising Diffusion Probabilistic Models (DDPM) (Ho et al., 2020; Song et al., 2020) were introduced as a probabilistic framework for image generation, achieving state-of-the-art (SOTA) performance in this domain. The core idea of DDPM is to model the process of gradually adding noise to a clean image, eventually reaching pure noise. The model learns to reverse process: starting from noisy data, it attempts to recover the original clean image at each timestep. To generate new images, DDPM samples random noise from a learned distribution and sequentially denoises it through the reverse process, ultimately producing a high-quality image. Over time, this method has been successfully extended to a wide range of computer vision tasks, demonstrating its versatility and effectiveness.

Building on previous work, we propose DiffuDETR, a framework built upon DeformableDETR and DiffuDINO, built on DINO, which shares DETR's backbone that extracts multi-scale image features, an encoder that employs multi-scale deformable attention within its transformer layers, a transformer decoder that applies cross-attention between initial object queries and the encoder's image features, and an MLP head that decodes each object query into a class label and bounding-box coordinates. We propose a new query initialization technique that aligns with the objective of denoising diffusion models to sample from the normally distributed reference points. This method avoids the inconveniences of query reference points initialization in DETR variants. Figure 1 shows the convergence of our proposed DiffuDINO against other DETR-based models, while it required more epochs due to the slow nature of training diffusion models. The performance surpasses DINO after 50 epochs of training on COCO dataset, which tends to deteriorate after 36 epochs.

We summarize our contributions in the following:

1. We represent object detection with detection transformers as a diffusion denoising process by denoising queries' reference points.

2. We introduce two models, DiffuDETR and DiffuDINO, built upon Deformable DETR and DINO, respectively.

3. We conduct extensive experiments with our models on multiple benchmark datasets and conduct extensive ablation studies to validate their effectiveness.

## 2 RELATED WORK

### 2.1 DIFFUSION MODELS

Denoising diffusion models have emerged as powerful generative frameworks in computer vision, demonstrating exceptional performance in tasks such as image generation (Austin et al., 2021; Avrahami et al., 2022), super-resolution (Gao et al., 2023), inpainting (Lugmayr et al., 2022), and editing. These models progressively learn to reverse a diffusion process that adds noise to data, enabling them to synthesize high-quality and diverse samples.

Beyond generative tasks, diffusion models have also proven useful in discriminative settings, including image segmentation (Amit et al., 2021; Baranchuk et al., 2021; Brempong et al., 2022), classification (Chen et al., 2023a), and anomaly detection (Wolleb et al., 2022). Their ability to learn strong latent representations highlights their potential in broader representation learning contexts, making them an increasingly popular choice across multiple vision domains (Croitoru et al., 2023).

While diffusion models have seen significant success in generative and discriminative tasks, their application to object detection remains relatively underexplored. Only a few works (Chen et al., 2023b) have investigated generative diffusion models for detection, and the progress in this area noticeably lags behind that in segmentation. This discrepancy arises because segmentation tasks are naturally formulated in an image-to-image fashion, which aligns closely with the denoising and generative process of diffusion models. In contrast, object detection is inherently a set prediction problem, requiring the model to generate a discrete set of object candidates and assign them to corresponding ground truth objects (Ren et al., 2015). This difference introduces unique challenges for diffusion-based approaches, as the generation of unordered object queries and accurate localization is conceptually less straightforward than reconstructing pixel-wise maps.

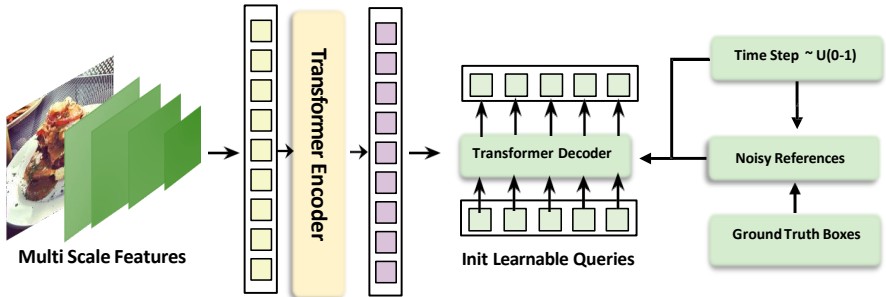

Figure 2: The DiffuDETR model framework involves a transformer encoder that encodes multi-scale visual features from the backbone. Ground-truth object reference points are introduced with Gaussian noise, where the noise level is controlled by a time step. These noisy reference points, along with learnable content queries, are processed by the transformer decoder, which denoises them based on the encoded features to produce precise object localizations during inference.

## 2.2 DETECTION TRANSFORMER

DEtection TRansformer (DETR) (Carion et al., 2020) revolutionized object detection by reformulating it as a direct set prediction problem, eliminating the need for hand-crafted components such as anchor boxes, region proposals, or non-maximum suppression. At its core, DETR employs an encoder-decoder transformer architecture: the encoder processes image features extracted by a CNN backbone, while the decoder operates on a fixed set of learnable object queries. Each query acts as a placeholder for a potential object and interacts with encoded features through cross-attention. Training is performed end-to-end using a bipartite matching loss based on the Hungarian algorithm, which enforces one-to-one alignment between predictions and ground-truth objects. This design enables DETR to predict all objects in a single forward pass, making it both conceptually simple and architecture-agnostic.

Despite its conceptual clarity, DETR faces practical challenges. Since its queries are initialized as zero embedding vectors without explicit spatial priors, the model must learn query-to-object alignment entirely from scratch, leading to slow convergence and unstable training. These limitations have motivated two complementary directions of research: improving query initialization and designing more effective training objectives.

The first direction focuses on enhancing query representation and initialization. DAB-DETR (Dynamic Anchor Box DETR) (Liu et al., 2022) reformulates decoder queries explicitly as anchor box coordinates—center, width, and height—which are dynamically updated across decoder layers. By incorporating positional priors directly into the queries, DAB-DETR improves alignment with image features and refines localization progressively, resulting in faster convergence and higher accuracy. In parallel, Deformable DETR (Zhu et al., 2020) introduces Deformable Attention, which restricts attention to a sparse set of spatially relevant sampling points, allowing the model to focus on key object regions. Its two-stage variant further strengthens query content by generating region proposals in the encoder (topK proposals) and refining them in the decoder, making queries image-dependent. Together, these works demonstrate that better-designed queries with spatial priors and dynamic updates can significantly enhance convergence and detection performance.

The second line of research focuses on auxiliary training tasks that enhance optimization and stability. DN-DETR (Li et al., 2022) introduces a denoising auxiliary task, where noisy ground-truth boxes and labels are injected during training, and the model learns to reconstruct the correct targets. This additional supervision alleviates instability in bipartite matching and accelerates convergence. Building upon this idea, DINO (Zhang et al., 2022) incorporates a contrastive denoising (CDN) auxiliary task combined with mixed query selection. Reference points are sampled from high-confidence encoder outputs, while their content queries are initialized with learnable class embeddings, aligning better with the denoising objective. Furthermore, DINO introduces hard nega-

tives to strengthen the auxiliary contrastive loss, yielding improved robustness and overall detection performance.

Building on these advances, we introduce DiffuDETR, a diffusion-based object detector that leverages a generative denoising process to produce object query anchors from noise. By progressively denoising noisy reference points, DiffuDETR generates better-initialized queries that provide strong starting points for the decoder, effectively addressing limitations in query alignment and convergence. At the same time, the denoising process serves as an improved training objective, guiding the model to recover precise object locations and class predictions. This dual advantage of enhanced query anchor initialization and a denoising-driven learning signal allows DiffuDETR to achieve greater training stability and superior detection performance compared to existing DETR variants.

Beyond these two research directions, several additional approaches have explored alternative pathways for improving DETR-based detectors. Many-to-one supervision strategies (Zhao et al., 2024; Ouyang-Zhang et al., 2022) relax the strict one-to-one matching constraint to provide richer supervisory signals. Other works introduce multi-route decoding to enhance information flow (Zhang et al., 2025) and improve optimization. Aside from architectural changes, loss-function refinements aim to better align classification and localization objectives (Cai et al., 2024). While these directions offer promising directions for advancing DETR-style models, they fall outside the scope of this paper, which focuses on diffusion-based query initialization and denoising-driven training.

## 2.3 Object Detection As Generation Task

Recent studies have started to view object detection through the lens of generative modeling, moving beyond traditional discriminative formulations. In this perspective, detection is cast as the process of generating structured outputs (bounding boxes and class labels) conditioned on image features. Two notable directions are sequence generation and denoising diffusion.

Pix2Seq (Chen et al., 2021) is one of the first works to explore this paradigm, formulating detection as a sequence generation task. Bounding boxes and class labels are represented as discrete tokens, and an encoder–decoder architecture is trained to autoregressively generate these tokens conditioned on the image and previously generated outputs. By doing so, Pix2Seq eliminates the need for hand-crafted components such as proposal generation or bounding box regression, offering a simple, generic formulation of detection.

In contrast, DiffusionDet (Chen et al., 2023b) formulates detection as a denoising diffusion process. Built on the Sparse R-CNN decoder (Sun et al., 2021), DiffusionDet replaces fixed proposal boxes with noisy ones, which are progressively refined into accurate predictions.

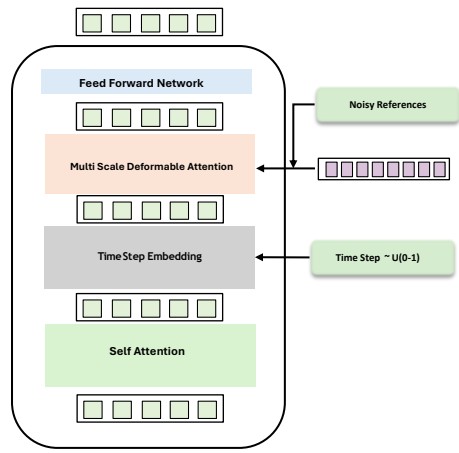

Figure 3: DiffuDETR's decoder iteratively denoises noisy object reference points using multi-scale encoded features, integrating self-attention, deformable attention, feed-forward networks, and time-step embeddings to refine queries across diffusion steps.

During training, ground-truth boxes are diffused into random distributions, and the model learns to reverse this noising process. At inference, boxes sampled from a Gaussian distribution are iteratively denoised through multiple cascaded decoder stages, enabling progressive refinement and flexibility in the number of predictions.

While Pix2Seq treats detection as language modeling and DiffusionDet as denoising noisy proposal boxes, our work DiffuDETR continues along the diffusion direction but adapts it to DETR. Specifically, we formulate object detection as a denoising diffusion process that generates DETR's object queries' anchors directly from noise, introducing diffusion-based query generation into transformer-based detection.

# 3 METHOD

## 3.1 PRELIMINARIES

### 3.1.1 DETR

The DETR model Carion et al. (2020) uses an image feature extracted as a backbone for feature extraction, followed by a Transformer encoder that processes the feature map into sequences. The decoder, which utilizes learned object queries, predicts bounding boxes and object classes. The Hungarian Matcher is used to match predicted boxes with ground truth by minimizing a cost matrix based on class and bounding box errors. Deformable DETR (Zhu et al., 2020) introduced deformable attention, where each query generates reference points to facilitate more efficient computations. This approach allows the queries to leverage multi-scale features, improving object localization. Additionally, Deformable DETR introduced a 2-stage initialization method, where queries are initialized with the top-k encoder proposals.

### 3.1.2 DENOISING DIFFUSION PROBABILISTIC MODELS

In Denoising Diffusion Probabilistic Models (DDPM), the forward process refers to the gradual addition of noise to an image over a series of timesteps, which transforms a clean image $x_0$ into a noisy vector $x_t$. This process is performed in a series of steps, each adding a small amount of Gaussian noise to the image, progressively degrading it. Mathematically, this forward diffusion process is defined as:

$$q(x_t|x_{t-1}) = \mathcal{N}(x_t; \sqrt{1-\beta_t}x_{t-1}, \beta_t I) \tag{1}$$

where $\beta_t$ is the noise scheduler to control the mean of the added noise, and $\mathcal{N}$ is the normal distribution of the added noise.

During sampling, we can generate samples by sampling a noisy image $x_T \sim \mathcal{N}(0, I)$, we update the noisy image using the following equation:

$$x_{t-1} = \frac{1}{\sqrt{\alpha_t}} \left( x_t - \frac{\sqrt{1-\alpha_t}}{1-\bar{\alpha}_t} \right) \epsilon_\theta(x_t, t, y) + \sigma_t.z \tag{2}$$

where $\alpha_t = 1 - \beta_t$ and $\bar{\alpha}_t = \prod_{s=1}^{T} \alpha_s$ and $\beta_t$ is scheduled to control the mean of the noise added to the original image. $z \sim \mathcal{N}(0, I)$ and $\sigma_t$ are used to control the stochasticity of sampling.

## 3.2 DIFFUDETR

Our models build upon the Deformable DETR framework, retaining the encoder-decoder transformer architecture with multi-scale deformable attention. DiffuDETR is built on Deformable-DETR and DiffuDINO adds the CDNs introduced in DINO (Zhang et al., 2022). The key modification is introduced at the training scheme. We adopt a diffusion-like training where reference points are treated as low-dimensional latent diffusion variables and refined through an iterative denoising process to enhance object localization. During training, we use a diffusion schedule with 100 timesteps $T = 100$, which is significantly fewer than the typical 1000 steps $T = 1000$ used in standard diffusion models. This reduction is made possible by the low dimensionality of the diffusion space, which allows efficient sampling during inference.

The complete architecture is illustrated in Figure 2. The encoder is similar to that introduced in Zhu et al. (2020), which extracts multi-scale visual features from the input image using deformable attention modules, effectively capturing rich contextual information at various spatial resolutions. These encoded features serve as the foundational representation for the detection pipeline. The transformer decoder layer shown in Figure 3 takes three inputs: the encoded multi-scale features $O_{\text{enc}}$, the noisy reference points $r_t$, and static learnable content queries. While the content queries encode semantic information about potential object classes, the noisy reference points provide spatial priors that guide the denoising process. The decoder iteratively refines these noisy queries conditioned on the encoded image features, effectively denoising and localizing objects.

$$q_n = \text{FFN}(\text{MSDA}(\text{SA}(q_{n-1}) + t), r_t, O_{enc}) \tag{3}$$

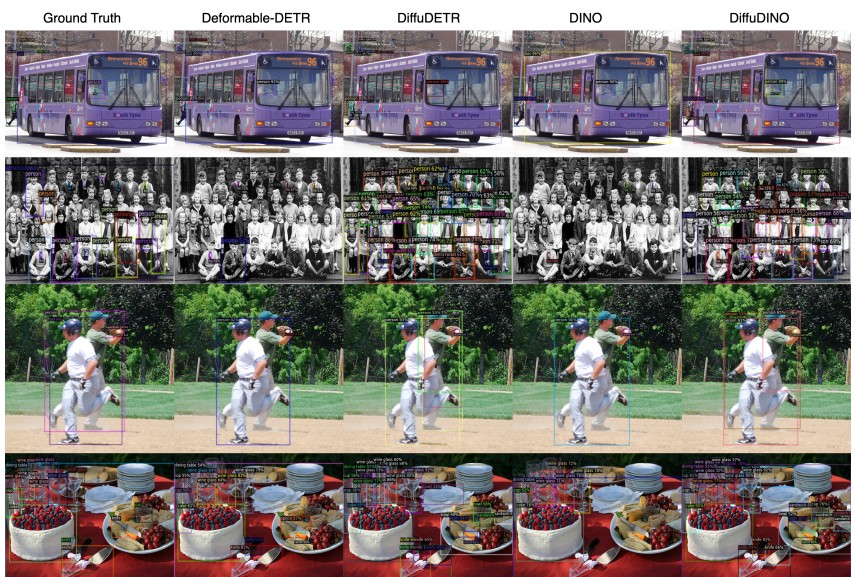

Figure 4: Qualitative comparison between Deformable DETR, DiffuDETR, DINO, and DiffuDINO on COCO 2017 Validation set. Only predictions with confidence scores above 50% are shown.

where $q_n$ represents the $n^{th}$ layer queries, $t$ represents the timestep embedding, $O_{enc}$ is the output multi scale features of the encoder. SA stands for self-attention, MSDA stands for Multi-scale deformable attention that takes the noisy reference points and interpolates the sampled points on the encoded features.

We introduce the query denoising task as simply a diffusion process on the normalized object reference points $r \in \mathbb{R}^{N \times 4}$, representing $N$ ground-truth objects in an image. Specifically, we sample initial time step $t \sim \mathcal{U}(0, 100)$ reference points $r_t$ from a posterior Noise distribution given by:

$$q(r_t|r) = f(r_t; r, \sigma^2 I), \tag{4}$$

where $r_t$ are noisy reference points at diffusion step $t$, and $\sigma^2$ denotes the variance controlling the noise intensity. This formulation encourages the model to learn the conditional distribution of object locations. $f$ stands for the forward process refers to the noise function used for noising conveniently chosen to be a normal distribution; however, more options could suit different problems (Nachmani et al., 2021).

At inference time, we generate object proposals by applying a deterministic DDIM sampler over a small number of timesteps. Concretely, for $N$ object queries, we first sample $K = 4$ reference points per query from a standard Gaussian at the final diffusion step:

$$r_T^{(i,k)} \sim \mathcal{N}(0, I), \quad i = 1, \dots, N, \quad k = 1, \dots, 4.$$

We then perform iterative denoising for $t = T, T - 1, \dots, 1$. At each step, we predict the noise residual $\hat{\epsilon} = \epsilon_\theta(r_t, t)$ via the transformer decoder and update the reference points according to the DDIM update rule (Song et al., 2020):

$$r_{t-1} = \sqrt{\bar{\alpha}_{t-1}} \, \frac{r_t - \sqrt{1 - \bar{\alpha}_t} \, \hat{\epsilon}}{\sqrt{\bar{\alpha}_t}} \; + \; \sqrt{1 - \bar{\alpha}_{t-1}} \, \hat{\epsilon}, \tag{5}$$

Different noise schedulers have been proposed for diffusion models (Chen, 2023), with performance varying across tasks. In our setting, inference requires only $S$ decoding evaluations, where $S \ll T$. The outputs $\{r_0^{(i,k)}\}$ from these steps are taken as the final bounding boxes. This efficient sampling scheme adds only a few extra computations using the decoder while the backbone and encoder are run once, resulting in only a small increase in computation (GFLOPs) compared to DINO.

Table 1: Comparison of various object detectors on the COCO 2017 validation set using ResNet-50 and ResNet-101 backbones.

| Model | Epochs | AP | $AP_{50}$ | $AP_{75}$ | $AP_s$ | $AP_m$ | $AP_l$ |
|---|---|---|---|---|---|---|---|
| **ResNet-50** (He et al., 2016) | | | | | | | |
| DETR-DC5 (Carion et al., 2020) | 500 | 43.3 | 63.1 | 45.9 | 22.5 | 47.3 | 61.1 |
| DN-Deformable DETR (Li et al., 2022) | 50 | 48.6 | 67.4 | 52.7 | 31.0 | 52.0 | 63.7 |
| MS-DETR (Zhao et al., 2024) | 24 | 50.9 | 68.4 | 56.1 | 34.7 | 54.3 | 65.1 |
| Salience DETR(Hou et al., 2024) | 24 | 51.2 | 68.9 | 55.7 | 33.9 | 55.5 | 65.6 |
| MR-DETR (Zhang et al., 2025) | 24 | 51.4 | 69.0 | 56.2 | 34.9 | 54.8 | 66.0 |
| Pix2Seq (Chen et al., 2021) | 300 | 43.2 | 61.0 | 46.1 | 26.6 | 47.0 | 58.6 |
| DiffusionDet (Chen et al., 2023b) | - | 46.8 | 65.3 | 51.8 | 29.6 | 49.3 | 62.2 |
| Deformable DETR (Zhu et al., 2020) | 50 | 48.2 | 67.0 | 52.2 | 30.7 | 51.4 | 63.0 |
| Align-DETR (Cai et al., 2024) | 24 | 51.4 | 69.1 | 55.8 | 35.5 | 54.6 | 65.7 |
| DINO (Zhang et al., 2022) | 36 | 50.9 | 69.0 | 55.3 | 34.6 | 54.1 | 64.6 |
| DiffuDETR (Ours) | 50 | 50.2 | 66.8 | 55.2 | 33.3 | 53.9 | 65.8 |
| DiffuAlignDETR (Ours) | 24 | **51.9** | 69.2 | **56.4** | 34.9 | 55.6 | 66.2 |
| DiffuDINO (Ours) | 50 | **51.9** | 69.4 | 55.7 | **35.8** | 55.7 | **67.1** |
| **ResNet-101** (He et al., 2016) | | | | | | | |
| DETR-DC5 (Carion et al., 2020) | 50 | 43.5 | 63.8 | 46.4 | 21.9 | 48.0 | 61.8 |
| DAB-DETR-DC5 (Liu et al., 2022) | 50 | 46.6 | 67.0 | 50.2 | 28.1 | 50.5 | 64.1 |
| DN-DETR-DC5 (Li et al., 2022) | 50 | 47.3 | 67.5 | 50.8 | 28.6 | 51.5 | 65.0 |
| MR-DETR (Zhang et al., 2025) | 12 | 51.4 | 68.6 | 55.7 | **34.3** | 55.1 | 66.7 |
| Pix2Seq (Chen et al., 2021) | 300 | 44.5 | 62.8 | 47.5 | 26.0 | 48.2 | 60.3 |
| DiffusionDet (Chen et al., 2023b) | - | 47.5 | 65.7 | 52.0 | 30.8 | 50.4 | 63.1 |
| DINO (Zhang et al., 2022) | 12 | 50.0 | 67.7 | 54.4 | 32.2 | 53.4 | 64.3 |
| Align-DETR (Cai et al., 2024) | 12 | 51.2 | 68.8 | 55.7 | 32.9 | 55.1 | 66.6 |
| DiffuDINO (Ours) | 12 | 51.2 | 68.6 | 55.8 | 33.2 | **55.6** | **67.2** |
| DiffuAlignDETR (Ours) | 12 | **51.7** | **69.3** | **56.1** | 34.0 | **55.6** | 67.0 |

## 4 EXPERIMENTS

### 4.1 SETUP

We evaluated DiffuDETR on multiple benchmark datasets to assess its performance across varying levels of object density, diversity, and scale.

**COCO 2017**. The COCO 2017 dataset (Lin et al., 2014) comprises 80 object categories, with 118,287 training images and 5,000 validation images. It serves as a standard benchmark for object detection, segmentation, and captioning tasks, featuring a diverse range of everyday scenes.

**LVIS**. The LVIS dataset (Gupta et al., 2019) includes 1,203 object categories, with 100,170 training images and 19,809 validation images. It is designed to address long-tail object distributions, featuring a large vocabulary of object categories with varying frequencies.

**V3DET**. The V3DET dataset (Wang et al., 2023) encompasses 13,204 object categories, with 183,354 training images and 29,821 validation images. It is a large-scale, richly annotated dataset featuring detection bounding box annotations for a vast number of object classes.

For evaluation, we adopt the standard metrics defined by each benchmark. On COCO, we report mean Average Precision (AP) averaged across IoU thresholds from 0.5 to 0.95, together with AP at IoU thresholds of 0.5 ($AP_{50}$) and 0.75 ($AP_{75}$), as well as scale-specific AP for small, medium, and large objects. On LVIS, we follow its official evaluation protocols, which extend the COCO-style AP by also reporting performance across categories with different frequencies: rare ($AP_r$), common ($AP_c$), and frequent ($AP_f$). These standardized measures provide a comprehensive view of detector performance across imbalanced category distributions.

### 4.2 RESULTS

Table 1 compares our proposed models with existing detectors on the COCO 2017 validation set. Using ResNet-50, DiffuDETR improves upon its baseline Deformable DETR by achieving 50.2

Table 2: Comparison of DINO and DiffuDINO on the LVIS validation set using ResNet-50 and ResNet-101 backbones. All models are trained for 12 epochs.

| Model | AP | $AP_{50}$ | $AP_{75}$ | $AP_s$ | $AP_m$ | $AP_l$ | $AP_r$ | $AP_c$ | $AP_f$ |
|---|---|---|---|---|---|---|---|---|---|
| **ResNet-50** (He et al., 2016) | | | | | | | | | |
| DINO (Zhang et al., 2022) | 26.5 | 35.9 | 27.8 | 20.0 | 35.2 | 40.9 | 9.2 | 24.6 | 36.2 |
| DiffuDINO (Ours) | **28.9** | **38.5** | **30.8** | **20.7** | **37.5** | **46.4** | **13.7** | **27.6** | **36.9** |
| **ResNet-101** (He et al., 2016) | | | | | | | | | |
| DINO (Zhang et al., 2022) | 30.9 | 40.4 | 32.8 | 23.2 | 40.5 | 46.3 | **13.9** | 29.7 | 39.7 |
| DiffuDINO (Ours) | **32.5** | **42.4** | **34.8** | **23.5** | **43.4** | **49.7** | 13.5 | **32.0** | **41.5** |

Table 3: Comparison of DINO and DiffuDINO on the V3DET validation set using ResNet-50 and Swin-B backbones. All models are trained for 24 epochs.

| Model | AP | $AP_{50}$ | $AP_{75}$ |
|---|---|---|---|
| **ResNet-50** (He et al., 2016) | | | |
| DINO (Zhang et al., 2022) | 33.5 | 37.7 | 35.0 |
| DiffuDINO (Ours) | **35.7** | **41.4** | **37.7** |
| **Swin-B** (Liu et al., 2021) | | | |
| DINO (Zhang et al., 2022) | 42.0 | 46.8 | 43.9 |
| DiffuDINO (Ours) | **50.3** | **56.6** | **52.9** |

Table 4: DiffuDINO performance with different diffusion noise distributions on the COCO 2017 validation set.

| Diffusion Noise | AP | $AP_{50}$ | $AP_{75}$ |
|---|---|---|---|
| Beta | 49.5 | 66.7 | 53.8 |
| Sigmoid Gaussian | 50.4 | 68.0 | 54.7 |
| Gaussian | **51.9** | **69.5** | **56.3** |

AP compared to 48.2, while also outperforming DN-Deformable DETR (48.6 AP), showing that our diffusion-based denoising task provides a stronger training signal than DN-DETR denoising strategy. Similarly, DiffuDINO surpasses its baseline DINO, reaching 51.9 AP versus 50.9, with consistent improvements across $AP_{50}$, $AP_{75}$, and scale-specific metrics. When compared with other generation-inspired approaches, both DiffuDETR and DiffuDINO significantly outperform DiffusionDet (46.8 AP) and Pix2Seq (43.2 AP), highlighting the advantages of integrating the diffusion denoising process with DETR-style query generation. With ResNet-101, DiffuDINO further improves over DINO (51.2 vs. 50.0 AP), demonstrating that our approach consistently strengthens DETR-based detectors across different backbones. We additionally introduce DiffuAlignDETR, built upon Align-DETR Cai et al. (2024), and observe that diffusion refinement provides a consistent boost under standard COCO training schedules. With a ResNet-50 backbone and the 2× schedule (24 epochs), Align-DETR achieves 51.4 AP, whereas DiffuAlignDETR improves this to 51.9 AP. Similarly, with ResNet-101 under the 1× schedule (12 epochs), DiffuAlignDETR reaches 51.7 AP compared to 51.2 AP for the baseline.

We show the results on LVIS validation set in Table 2. where we compare DiffuDINO with its baseline DINO under both ResNet-50 and ResNet-101 backbones, trained for 12 epochs. With ResNet-50, DiffuDINO achieves 28.9 AP, improving over DINO's 26.5 by +2.4 AP. With ResNet-101, DiffuDINO continues to outperform DINO (32.5 vs. 30.9 AP), with improvements in medium and large objects as well as common and frequent categories.

We present results on the V3DET validation set with ResNet-50 and Swin-B backbones in Table 3, trained for 24 epochs. With ResNet-50, DiffuDINO surpasses its baseline DINO by +2.2 AP (35.7 vs. 33.5), alongside clear improvements in $AP_{50}$ (+3.7) and $AP_{75}$ (+2.7). The performance gap becomes more pronounced with the stronger Swin-B backbone, where DiffuDINO achieves 50.3 AP, outperforming DINO's 42.0 by a large margin of +8.3 points.

Figure 4 visually compares detection results across four models: Deformable DETR, DiffuDETR, DINO, and DiffuDINO on samples from COCO 2017 Validation. The qualitative examples highlight the consistent improvements introduced by our diffusion-based models, particularly in crowded scenes with multiple overlapping objects. DiffuDETR exhibits more accurate and complete localization of instances compared to Deformable DETR, while DiffuDINO further refines predictions beyond DINO, reducing missed detections and improving boundary alignment. These visualizations support the quantitative results and confirm that the diffusion-based denoising process yields clearer and more precise predictions, especially in challenging, densely populated regions.

### 4.3 ABLATION STUDY

#### 4.3.1 DIFFUSION NOISE DISTRIBUTIONS

Table 4 reports the results using different noise distributions. Among the tested, Gaussian distribution consistently achieves the best performance across all metrics, yielding 51.9 AP. In comparison, Beta noise distribution underperforms relative to Gaussian noise, with up to 2.4 AP lower. Sigmoid Gaussian shows competitive results. Our primary intuition for using the sigmoid Gaussian distribution is to avoid clipping values from the diffused Gaussian points (ensuring that reference points are valued between (0 and 1). However, it still lags behind Gaussian noise overall. This shows that empirically Gaussian distribution still prevails as the more suitable distribution even in detection tasks.

Table 5: DiffuDINO performance with different noise schedulers on the COCO 2017 validation set.

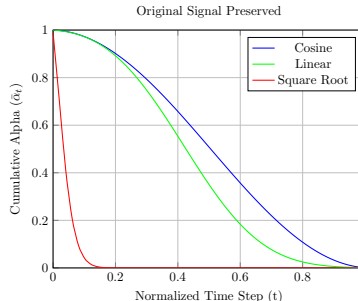

Figure 5: Different Schedulers noise retained plot against timesteps.

| Scheduler | AP | $AP_{50}$ | $AP_{75}$ | $AP_s$ | $AP_m$ | $AP_l$ |
|---|---|---|---|---|---|---|
| Cosine | **51.9** | **69.5** | **56.4** | **35.9** | **55.8** | **67.1** |
| Linear | 51.6 | 69.1 | 56.2 | 35.6 | **55.8** | 66.4 |
| Square Root | 51.4 | 68.9 | 56.0 | 35.3 | 55.4 | 66.0 |

Table 6: Comparison of DINO and DiffuDINO with different Decoder Evaluation on the COCO 2017 validation set. We report AP metrics along with FLOPs (in GigaFLOPs) and activations (in millions).

| Model | D.E. | AP | FLOPs (G) | Activations (M) |
|---|---|---|---|---|
| DINO | 1 | 50.9 | $244.5 \pm 25.5$ | $673.1 \pm 66.5$ |
| DiffuDINO | 1 | 51.6 | $244.5 \pm 25.5$ | $673.1 \pm 66.5$ |
| DiffuDINO | 3 | 51.9 | $285.2 \pm 27.1$ | $871.0 \pm 72.7$ |
| DiffuDINO | 5 | 51.8 | $326.0 \pm 28.7$ | $1068.9 \pm 79.0$ |
| DiffuDINO | 10 | 51.4 | $427.9 \pm 32.7$ | $1563.6 \pm 94.7$ |

#### 4.3.2 DIFFUSION NOISE SCHEDULERS

We compare the impact of different noise schedulers on DiffuDINO performance in Table 5. The cosine scheduler achieves the best overall results, reaching 51.9 AP. The linear scheduler performs competitively, particularly on medium-sized objects where it matches cosine at 55.8 $AP_m$, but slightly trails behind in other metrics. The square root scheduler shows the lowest performance among the three, with a drop of around 0.5 AP compared to cosine. This is expected because cosine schedulers tend to retain more of the original signal in later timesteps as shown in Figure 5. These results indicate that cosine scheduling provides a smoother distortion for reference points as time steps $t < 100$ while still completely distorting the reference points at $t = 100$.

#### 4.3.3 NUMBER OF DECODER EVALUATIONS

Table 6 shows the effect of varying decoder evaluation on DiffuDINO performance compared to the DINO baseline on the COCO 2017 validation set. With a single decoder evaluation, DiffuDINO already surpasses DINO by +1.7 AP while maintaining identical computational cost in both FLOPs ($244.5 \pm 25.5$ G) and activations ($673 \pm 66$ M). Increasing the number of decoder evaluations to three yields the best overall results, reaching 51.9 AP with improvements across all object scales. However, this comes with a moderate increase in computation to $285.2 \pm 27.1$ GFLOPs and $871 \pm 73$ M activations. Further increasing the decoder evaluations to five or ten does not yield additional performance gains, with AP slightly declining while computational costs continue to grow significantly, reaching $428 \pm 33$ GFLOPs and $1564 \pm 95$ M activations at ten steps. This demonstrates that our approach enables effective sampling, with only three decoder evaluations being sufficient to achieve the best results while adding minimal computation overhead compared to the baseline.

Table 7: DiffuDINO performance with different numbers of decoder evaluations on COCO val2017, averaged over 5 random initializations. Results are reported as mean $\pm$ standard deviation.

| D.E. | AP | $AP_{50}$ | $AP_{75}$ | $AP_s$ | $AP_m$ | $AP_l$ |
|---|---|---|---|---|---|---|
| 1 | $51.68 \pm 0.02$ | $69.28 \pm 0.02$ | $55.89 \pm 0.04$ | $35.50 \pm 0.09$ | $55.58 \pm 0.04$ | $66.96 \pm 0.10$ |
| 3 | $51.95 \pm 0.03$ | $69.54 \pm 0.02$ | $56.32 \pm 0.05$ | $35.92 \pm 0.08$ | $55.81 \pm 0.01$ | $67.18 \pm 0.05$ |
| 5 | $51.83 \pm 0.01$ | $69.24 \pm 0.03$ | $56.21 \pm 0.05$ | $35.82 \pm 0.06$ | $55.71 \pm 0.06$ | $67.02 \pm 0.05$ |
| 10 | $51.49 \pm 0.08$ | $68.54 \pm 0.09$ | $56.02 \pm 0.08$ | $35.56 \pm 0.12$ | $55.46 \pm 0.15$ | $66.83 \pm 0.04$ |

Table 8: Comparison of DiffuDINO performance across varying numbers of decoder evaluations on COCO Dense and Sparse scene subsets. The first row of each section reports the baseline DINO results, followed by DiffuDINO results using different numbers of decoder evaluations. Metrics are reported as mean $\pm$ standard deviation over 5 random initializations.

| D.E. | AP | AP50 | AP75 | APS | APM | APL |
|---|---|---|---|---|---|---|
| | | | **COCO Sparse Scenes** | | | |
| DINO | 57.00 | 73.37 | 62.63 | 36.14 | 55.95 | 59.26 |
| 1 | $58.48 \pm 0.05$ | $74.68 \pm 0.031$ | $63.53 \pm 0.02$ | $37.78 \pm 0.10$ | $57.66 \pm 0.08$ | $68.34 \pm 0.13$ |
| 3 | $58.65 \pm 0.03$ | $74.75 \pm 0.03$ | $63.85 \pm 0.04$ | $38.07 \pm 0.19$ | $57.81 \pm 0.05$ | $68.55 \pm 0.06$ |
| 5 | $58.50 \pm 0.04$ | $74.52 \pm 0.05$ | $63.71 \pm 0.08$ | $37.72 \pm 0.15$ | $57.57 \pm 0.14$ | $68.46 \pm 0.04$ |
| 10 | $58.16 \pm 0.04$ | $73.94 \pm 0.03$ | $63.41 \pm 0.07$ | $37.36 \pm 0.14$ | $57.27 \pm 0.15$ | $68.10 \pm 0.04$ |
| | | | **COCO Dense Scenes** | | | |
| DINO | 43.72 | 62.29 | 47.81 | 33.98 | 51.95 | 66.63 |
| 1 | $44.53 \pm 0.07$ | $63.24 \pm 0.08$ | $47.87 \pm 0.10$ | $34.45 \pm 0.11$ | $53.15 \pm 0.16$ | $61.48 \pm 0.16$ |
| 3 | $44.88 \pm 0.02$ | $63.65 \pm 0.03$ | $48.39 \pm 0.09$ | $34.96 \pm 0.09$ | $53.40 \pm 0.06$ | $61.60 \pm 0.06$ |
| 5 | $44.85 \pm 0.05$ | $63.36 \pm 0.09$ | $48.40 \pm 0.11$ | $34.92 \pm 0.07$ | $53.46 \pm 0.07$ | $61.61 \pm 0.07$ |
| 10 | $44.57 \pm 0.07$ | $62.69 \pm 0.10$ | $48.23 \pm 0.11$ | $34.79 \pm 0.15$ | $53.35 \pm 0.10$ | $61.31 \pm 0.09$ |

## 4.4 SENSITIVITY TO INITIALIZATION NOISE

To assess the robustness of DiffuDINO with respect to initial noise, we evaluate the model across five independent runs with different random seeds and report the mean and standard deviation of AP and related metrics. As shown in Tables 7, DiffuDINO exhibits consistently low seed-to-seed variance across all decoder evaluation settings (1, 3, 5, and 10 steps). On COCO with a ResNet-50 backbone, the variation across seeds remains below $\pm 0.2$ AP, demonstrating that the model's predictions are highly stable despite changes in initialization noise.

To further analyze stability under different scene complexities, we split the COCO validation set into a sparse scenes subset (images with 10 or fewer objects) and a dense scenes subset (images with more than 10 objects). DiffuDINO maintains the same level of robustness in both subsets. As shown in the Dense and Sparse results in Table 8, the standard deviation remains below $\pm 0.2$ AP across all metrics, even under large variations in object density. Moreover, DiffuDINO consistently improves over the baseline DINO in both crowded and sparse scenes, demonstrating that diffusion-based refinement not only enhances accuracy but also preserves stability across seeds.

## 5 CONCLUSION

In this work, we represented object detection with detection transformers as a diffusion denoising process by progressively denoising queries' reference points. We introduced two models, DiffuDETR and DiffuDINO, built upon Deformable DETR and DINO, respectively. Our approach enables effective sampling at inference, where only three decoder evaluations are sufficient to achieve the best results while adding minimal computation overhead compared to the baseline. Extensive experiments on COCO, LVIS, and V3Det demonstrate that our method consistently improves over the baselines across all datasets. Furthermore, we show that Gaussian noise provides the most suitable training signal, consistently outperforming alternative noise distributions. In addition, we observe that a cosine scheduler achieves the best performance among the different noise scheduling strategies we tested. Moreover, our multi-seed analysis confirms that DiffuDINO is highly robust to initialization noise. Finally, we believe this work opens up new directions for integrating generative and autoregressive approaches into object detection, offering fresh perspectives beyond traditional discriminative formulations.

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

# A   APPENDIX

## A.1   IMPLEMENTATION DETAILS

All experiments were implemented using the Detrex framework (Ren et al., 2023). For data pre-processing, we followed the default COCO augmentations provided by Detrex across all datasets to ensure consistent training pipelines. Unless otherwise stated, all models were optimized using the AdamW optimizer (Loshchilov & Hutter, 2017) with an initial learning rate of $1 \times 10^{-4}$, weight decay of $1 \times 10^{-4}$, and $\beta = (0.9, 0.999)$. For models trained with ResNet-50 and ResNet-101 backbones, we used a batch size of 16, while models with the Swin-B backbone were trained with a batch size of 8 due to memory constraints. Across all settings, we adopted four-scale feature maps from the backbone to ensure a consistent multi-scale representation for detection.

### A.1.1   TRAINING ON COCO 2017

For COCO 2017, we trained models with ResNet-50 and ResNet-101 backbones (He et al., 2016). Models with ResNet-50 backbones were trained for 50 epochs, applying a learning rate decay at epoch 40. Models with ResNet-101 backbones were trained for 12 epochs, with the learning rate decayed at epoch 11. We additionally retained the baseline DINO with ResNet-50 backbone for 50 epochs to provide a consistent reference. We found DINO after 36 epochs starts to deteriorate, so we compare our results to the best results achieved by the authors in their paper (Zhang et al., 2022).

### A.1.2   TRAINING ON LVIS

On the LVIS dataset, we trained both ResNet-50 and ResNet-101 backbones for 12 epochs, with a total of 270k iterations. The learning rate was decayed by factors of 0.1 and 0.01 at 210k and 250k iterations, respectively, before terminating at 270k iterations. To ensure comparability, we also trained the baseline DINO models with ResNet-50 and ResNet-101 backbones following the same schedule.

### A.1.3   TRAINING ON V3DET

For V3DET, we trained models with both ResNet-50 and Swin-B backbones (Liu et al., 2021) for 24 epochs, reducing the learning rate at epochs 16 and 22 by a factor of 10. A repeated sampler was used with the repeat factor set $1 \times 10^{-3}$. , following the dataset protocol. For baseline comparisons, we used the official DINO results reported in the V3DET dataset paper and replicated their hyperparameters in our DiffuDINO experiments to ensure fair comparison.

### A.1.4   DIFFUSION-BASED VARIANTS

All DETR-based models were trained with 900 object queries by default. Our DiffuDETR and DiffuDINO models introduced a diffusion process with 100 decoder evaluations, where gaussian noise was progressively denoised. During inference, we reduced this to only 3 denoising steps for efficiency. In addition, we adopted a cosine learning rate scheduler for diffusion-based models. We also conducted ablation studies to analyze the impact of different noise disturbances, diffusion schedulers, the number of queries, and inference decoder evaluations.

In the baseline DINO and DiffuDINO models, we employed 300 CDN denoising queries. For DiffuDETR, which is built upon Deformable DETR, we did not adopt the two-stage variant but instead incorporated the encoder loss to enhance representation learning. Furthermore, we applied exponential moving average (EMA) updates to both DiffuDETR and DiffuDINO, as EMA stabilizes training and improves convergence under the diffusion denoising objective.

## A.2 VISUALIZATION OF PREDICTIONS WITH DIFFERENT DECODER EVALUATION STEPS ON COCO

Figure 6 illustrates the effectiveness of DiffuDINO under varying numbers of decoder evaluation steps. Increasing the number of evaluations enables the model to detect more objects, particularly in complex and crowded scenes. Notably, even with a single evaluation step, DiffuDINO already surpasses the baseline DINO, demonstrating stronger localization and recall in challenging scenarios. This highlights the model's robustness and its ability to scale detection quality with additional decoding steps. The quantitative results are summarized in Table 9.

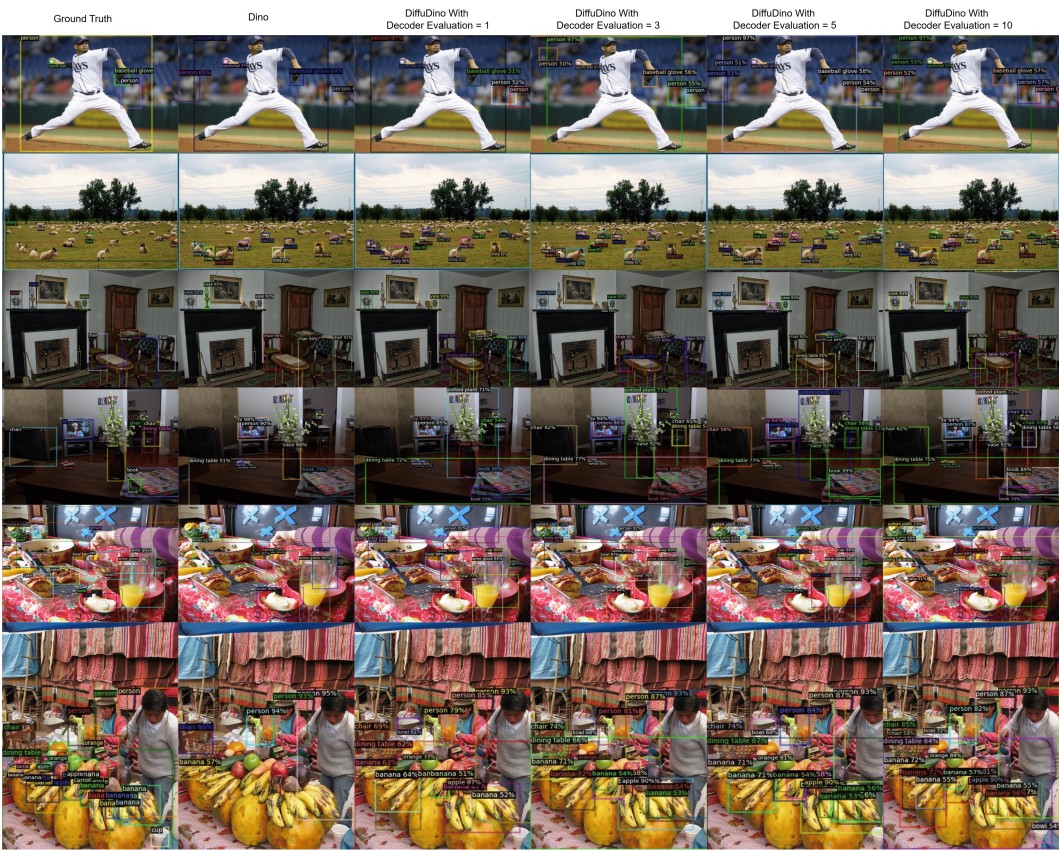

Figure 6: Qualitative comparison of object detection results on COCO 2017 validation set across different numbers of decoder evaluation steps ($t = 1, 3, 5, 10$), showing DiffuDINO, baseline DINO with ResNet-50 backbone, and ground truth annotations. Only predictions with confidence scores above 50% are shown.

Table 9: Comparison of DINO and DiffuDINO with different decoder evaluation steps on the COCO 2017 validation set.

| Model | Decoder Evaluation | AP | $AP_{50}$ | $AP_{75}$ | $AP_S$ | $AP_M$ | $AP_L$ |
|---|---|---|---|---|---|---|---|
| DINO | 1 | 50.9 | 69.0 | 55.3 | 34.3 | 54.1 | 64.6 |
| DiffuDINO | 1 | 51.6 | 69.2 | 55.8 | 35.5 | 55.5 | 66.9 |
| DiffuDINO | 3 | 51.9 | 69.5 | 56.3 | 35.9 | 55.8 | 67.1 |
| DiffuDINO | 5 | 51.8 | 69.2 | 56.2 | 35.8 | 55.7 | 67.0 |
| DiffuDINO | 10 | 51.4 | 68.5 | 56.0 | 35.5 | 55.4 | 66.8 |

A.3    VISUALIZATION OF PREDICTIONS ON LVIS

Figure 7 presents qualitative examples from the LVIS validation set, showing how DiffuDINO benefits from additional decoder evaluation steps. The model consistently produces more accurate detections in dense and diverse scenes, capturing objects that DINO often misses. Even at a single evaluation step, DiffuDINO demonstrates clear improvements over the baseline, highlighting its advantage in handling long-tail distributions and fine-grained categories. The corresponding quantitative results are provided in Table 10.

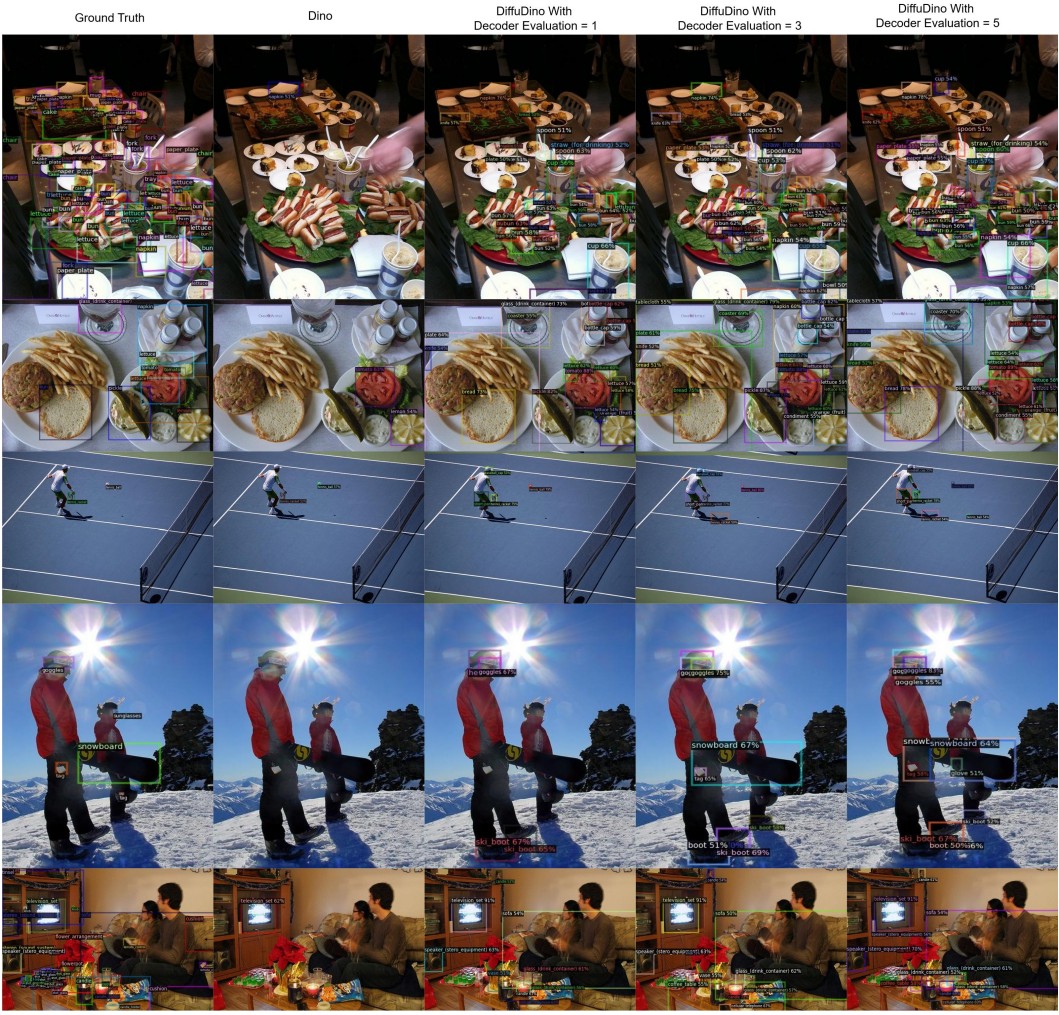

Figure 7: Qualitative comparison of object detection results on LVIS validation set across different numbers of decoder evaluation steps ($t = 1, 3, 5$), showing DiffuDINO, baseline DINO with ResNet-50 backbone, and ground truth annotations. Only predictions with confidence scores above 50% are shown.

Table 10: Comparison of DINO and DiffuDINO with different decoder evaluation steps on the LVIS validation set.

| Model | Decoder Evaluation | AP | $AP_{50}$ | $AP_{75}$ | $AP_s$ | $AP_m$ | $AP_l$ | $AP_r$ | $AP_c$ | $AP_f$ |
|---|---|---|---|---|---|---|---|---|---|---|
| DINO | 1 | 26.5 | 35.9 | 27.8 | 20.0 | 35.2 | 40.9 | 9.2 | 24.6 | 36.2 |
| DiffuDINO | 1 | 27.0 | 36.4 | 28.6 | 19.5 | 35.8 | 44.1 | 12.0 | 26.1 | 34.7 |
| DiffuDINO | 3 | 28.9 | 38.5 | 30.8 | 20.7 | 37.5 | 46.4 | 13.7 | 27.6 | 36.9 |
| DiffuDINO | 5 | 28.2 | 37.6 | 30.2 | 20.2 | 36.5 | 45.6 | 11.9 | 26.9 | 36.7 |

## A.4 NUMBER OF QUERIES

We ablate on the number of queries in Table 11 varying the number of queries on DiffuDINO performance on the COCO 2017 validation set, with all models trained under a linear noise scheduler. Increasing the number of queries from 300 to 900 yields noticeable improvements, particularly for small (35.6 $AP_s$) and medium (55.8 $AP_m$) objects. As expected, scaling more to 1500 queries provides the best overall AP (51.9), demonstrating that our approach continues to scale effectively with more queries. For consistency and fair comparison with baselines, however, all main experiments in this work are conducted with the conventional 900 queries.

Table 11: DiffuDINO performance with different numbers of queries on the COCO 2017 validation set. All models are trained with a linear noise scheduler.

| No. of Queries | AP | $AP_{50}$ | $AP_{75}$ | $AP_s$ | $AP_m$ | $AP_l$ |
|---|---|---|---|---|---|---|
| 300 | 51.2 | 68.8 | 55.6 | 35.0 | 55.2 | 66.4 |
| 900 | 51.6 | 69.1 | 56.2 | **35.6** | **55.8** | 66.4 |
| 1500 | **51.9** | **69.5** | **56.4** | 35.0 | 55.5 | **66.6** |

## A.5 NOISE DISTRIBUTIONS

In section 4.3.1 we ablate on different noise distributions. We use the following PDFs for sampling Gaussian Distribution

$$f_X(x) = \frac{1}{\sigma\sqrt{2\pi}} \exp\left(-\frac{(x-\mu)^2}{2\sigma^2}\right), \quad -\infty < x < \infty, \mu = 0.5, \sigma = 0.5. \tag{6}$$

and beta distribution

$$f_X(x) = \frac{1}{\mathrm{B}(\alpha,\beta)} x^{\alpha-1}(1-x)^{\beta-1}, \quad 0 < x < 1, \ \alpha = 2, \ \beta = 2. \tag{7}$$

Sigmoid Gaussian is simply

$$f_X(x) = sigmoid(\frac{1}{\sigma\sqrt{2\pi}} \exp\left(-\frac{(x-\mu)^2}{2\sigma^2}\right)), \quad -\infty < x < \infty, \mu = 0, \sigma = 1. \tag{8}$$

Notice that due to the nature of gaussian distribution, noise would need to be clipped to ensure that normalized reference points are between 0 and 1. Beta and Sigmoid Gaussian ensure that the limit is within the distribution itself and our chosen hyperparameters.

## A.6 COMPUTATIONAL COMPARISON OF DETR VARIANTS.

In Table 12, we provide a detailed comparison of computational requirements across several DETR-based detection models, all using a ResNet-50 backbone. We report both FLOPs (in GigaFLOPs) and activations (in millions). As shown, models that introduce denoising mechanisms such as DN DETR and DN Deformable DETR incur a noticeable increase in computation compared to their non-denoising counterparts. This trend is consistent with the broader progression of DETR variants: newer models typically achieve stronger performance by introducing architectural refinements or additional training signals, but at the cost of increased FLOPs and activation memory.

Within this landscape, DiffuDINO follows a predictable computational pattern. With a single decoder evaluation, its cost matches that of its baseline DINO. Increasing the number of denoising steps naturally increases computation, yet the growth is linear and controllable, allowing practitioners to balance accuracy and efficiency depending on deployment constraints. This extended comparison provides a clear view of the trade-offs across DETR variants and demonstrates that Diffu-DINO fits cleanly within the expected computational spectrum of modern DETR models.

Table 12: Compute cost comparison across detection models using ResNet-50. FLOPs are reported in GigaFLOPs and activations in millions (mean ± std).

| Model | D.E. | FLOPs (G) | Activations (M) |
|---|---|---|---|
| DETR | 1 | 83.7 ± 9.3 | 222.93 ± 24.25 |
| DN DETR | 1 | 89.2 ± 9.5 | 253.87 ± 26.10 |
| Deformable DETR | 1 | 171.3 ± 18.6 | 484.02 ± 52.16 |
| DN Deformable DETR | 1 | 231.3 ± 25.1 | 602.40 ± 64.72 |
| Align DETR | 1 | 244.5 ± 25.5 | 673.05 ± 66.46 |
| MR Deformable DETR | 1 | 258.0 ± 27.4 | 709.06 ± 73.95 |
| DINO | 1 | 244.5 ± 25.5 | 673.06 ± 66.46 |
| Diffu-DINO | 1 | 244.5 ± 25.5 | 673.06 ± 66.46 |
| Diffu-DINO | 3 | 285.2 ± 27.1 | 870.96 ± 72.74 |
| Diffu-DINO | 5 | 326.0 ± 28.7 | 1068.86 ± 79.01 |
| Diffu-DINO | 10 | 427.9 ± 32.7 | 1563.61 ± 94.70 |

Table 13: COCO results for DINO (baseline) and DiffuDINO across 5 seeds under varying numbers of decoder evaluations using ResNet-50 backbone.

| Model | D.E. | Seed | AP | AP50 | AP75 | APs | APm | APl |
|---|---|---|---|---|---|---|---|---|
| DINO | 1 | 1 | 50.58 | 68.41 | 55.46 | 34.26 | 53.85 | 65.25 |
| DiffuDINO | 1 | 1 | 51.64 | 69.25 | 55.83 | 35.38 | 55.60 | 66.92 |
| DiffuDINO | 1 | 2 | 51.71 | 69.28 | 55.94 | 35.64 | 55.52 | 67.08 |
| DiffuDINO | 1 | 3 | 51.68 | 69.28 | 55.87 | 35.53 | 55.59 | 66.80 |
| DiffuDINO | 1 | 4 | 51.70 | 69.33 | 55.89 | 35.51 | 55.63 | 67.00 |
| DiffuDINO | 1 | 5 | 51.68 | 69.29 | 55.93 | 35.47 | 55.60 | 67.02 |
| DiffuDINO | 3 | 1 | 51.97 | 69.55 | 56.30 | 35.85 | 55.81 | 67.21 |
| DiffuDINO | 3 | 2 | 51.98 | 69.55 | 56.36 | 35.95 | 55.84 | 67.26 |
| DiffuDINO | 3 | 3 | 51.95 | 69.52 | 56.36 | 35.93 | 55.79 | 67.11 |
| DiffuDINO | 3 | 4 | 51.96 | 69.58 | 56.36 | 36.06 | 55.82 | 67.21 |
| DiffuDINO | 3 | 5 | 51.90 | 69.51 | 56.23 | 35.84 | 55.80 | 67.14 |
| DiffuDINO | 5 | 1 | 51.85 | 69.28 | 56.25 | 35.86 | 55.65 | 66.97 |
| DiffuDINO | 5 | 2 | 51.82 | 69.22 | 56.11 | 35.93 | 55.66 | 67.08 |
| DiffuDINO | 5 | 3 | 51.85 | 69.29 | 56.24 | 35.78 | 55.77 | 67.07 |
| DiffuDINO | 5 | 4 | 51.84 | 69.21 | 56.26 | 35.77 | 55.79 | 66.99 |
| DiffuDINO | 5 | 5 | 51.81 | 69.21 | 56.20 | 35.79 | 55.69 | 66.99 |
| DiffuDINO | 10 | 1 | 51.40 | 68.43 | 55.97 | 35.53 | 55.29 | 66.82 |
| DiffuDINO | 10 | 2 | 51.60 | 68.68 | 56.11 | 35.44 | 55.65 | 66.85 |
| DiffuDINO | 10 | 3 | 51.56 | 68.58 | 56.13 | 35.60 | 55.59 | 66.85 |
| DiffuDINO | 10 | 4 | 51.48 | 68.54 | 55.98 | 35.76 | 55.36 | 66.86 |
| DiffuDINO | 10 | 5 | 51.45 | 68.48 | 55.95 | 35.50 | 55.42 | 66.76 |

## A.7 COMPREHENSIVE MULTI-SEED RESULT

To provide a complete view of model stability under different scene complexities and data distributions, we report full results for three evaluation settings: COCO Dense scenes, COCO Sparse scenes, and the full COCO validation set. The results are in Tables 13, 14, and 15, where the first row of each table contains the baseline DINO scores, followed by DiffuDINO performance under different numbers of decoder evaluations. Across all three tables, DiffuDINO consistently surpasses the baseline DINO model both in Dense scenes and Sparse scenes while exhibiting remarkably small seed-to-seed variability.

## A.8 USE OF LARGE LANGUAGE MODELS

Large Language Models (LLMs) were utilized for minor edits and revisions during the writing of this paper, primarily to correct grammar, spelling, and improve clarity in certain passages. No content was generated or substantively altered by the LLMs.

Table 14: COCO sparse scenes results for DINO (baseline) and DiffuDINO across 5 seeds under varying numbers of decoder evaluations using ResNet-50 backbone.

| Model | D.E. | seed | AP | AP50 | AP75 | APs | APm | APl |
|---|---|---|---|---|---|---|---|---|
| DINO | 1 | 1 | 57.00 | 73.38 | 62.64 | 36.14 | 55.90 | 59.26 |
| DiffuDINO | 1 | 1 | 58.47 | 74.70 | 63.51 | 37.91 | 57.62 | 68.22 |
| DiffuDINO | 1 | 2 | 58.52 | 74.70 | 63.52 | 37.69 | 57.73 | 68.35 |
| DiffuDINO | 1 | 3 | 58.42 | 74.66 | 63.51 | 37.87 | 57.59 | 68.23 |
| DiffuDINO | 1 | 4 | 58.47 | 74.65 | 63.55 | 37.68 | 57.60 | 68.40 |
| DiffuDINO | 1 | 5 | 58.56 | 74.72 | 63.57 | 37.76 | 57.78 | 68.54 |
| DiffuDINO | 3 | 1 | 58.65 | 74.75 | 63.86 | 38.10 | 57.91 | 68.53 |
| DiffuDINO | 3 | 2 | 58.64 | 74.72 | 63.91 | 37.77 | 57.80 | 68.59 |
| DiffuDINO | 3 | 3 | 58.72 | 74.77 | 63.89 | 38.31 | 57.78 | 68.63 |
| DiffuDINO | 3 | 4 | 58.63 | 74.80 | 63.81 | 38.08 | 57.78 | 68.46 |
| DiffuDINO | 3 | 5 | 58.65 | 74.73 | 63.80 | 38.11 | 57.79 | 68.55 |
| DiffuDINO | 5 | 1 | 58.57 | 74.57 | 63.82 | 37.81 | 57.64 | 68.45 |
| DiffuDINO | 5 | 2 | 58.50 | 74.56 | 63.66 | 37.92 | 57.48 | 68.42 |
| DiffuDINO | 5 | 3 | 58.44 | 74.45 | 63.61 | 37.51 | 57.53 | 68.44 |
| DiffuDINO | 5 | 4 | 58.52 | 74.55 | 63.78 | 37.65 | 57.79 | 68.48 |
| DiffuDINO | 5 | 5 | 58.48 | 74.49 | 63.72 | 37.73 | 57.43 | 68.54 |
| DiffuDINO | 10 | 1 | 58.15 | 73.95 | 63.33 | 37.54 | 57.27 | 68.03 |
| DiffuDINO | 10 | 2 | 58.18 | 73.96 | 63.35 | 37.30 | 57.05 | 68.13 |
| DiffuDINO | 10 | 3 | 58.22 | 73.99 | 63.49 | 37.42 | 57.43 | 68.10 |
| DiffuDINO | 10 | 4 | 58.18 | 73.90 | 63.41 | 37.40 | 57.40 | 68.14 |
| DiffuDINO | 10 | 5 | 58.11 | 73.92 | 63.48 | 37.14 | 57.22 | 68.13 |

Table 15: COCO dense scenes results for DINO (baseline) and DiffuDINO across 5 seeds under varying numbers of decoder evaluations using ResNet-50 backbone.

| Model | D.E. | seed | AP | AP50 | AP75 | APs | APm | APl |
|---|---|---|---|---|---|---|---|---|
| DINO | 1 | 1 | 43.72 | 62.30 | 47.81 | 33.99 | 51.96 | 66.63 |
| DiffuDINO | 1 | 1 | 44.47 | 63.22 | 47.77 | 34.30 | 53.26 | 61.47 |
| DiffuDINO | 1 | 2 | 44.51 | 63.25 | 47.81 | 34.46 | 53.00 | 61.48 |
| DiffuDINO | 1 | 3 | 44.64 | 63.29 | 47.99 | 34.50 | 53.36 | 61.24 |
| DiffuDINO | 1 | 4 | 44.48 | 63.12 | 47.82 | 34.38 | 52.98 | 61.68 |
| DiffuDINO | 1 | 5 | 44.59 | 63.34 | 47.98 | 34.61 | 53.16 | 61.58 |
| DiffuDINO | 3 | 1 | 44.88 | 63.71 | 48.53 | 35.01 | 53.40 | 61.54 |
| DiffuDINO | 3 | 2 | 44.93 | 63.65 | 48.44 | 34.88 | 53.48 | 61.62 |
| DiffuDINO | 3 | 3 | 44.86 | 63.63 | 48.30 | 35.02 | 53.31 | 61.63 |
| DiffuDINO | 3 | 4 | 44.88 | 63.65 | 48.37 | 34.85 | 53.43 | 61.69 |
| DiffuDINO | 3 | 5 | 44.89 | 63.66 | 48.31 | 35.07 | 53.39 | 61.55 |
| DiffuDINO | 5 | 1 | 44.83 | 63.36 | 48.29 | 34.83 | 53.40 | 61.63 |
| DiffuDINO | 5 | 2 | 44.77 | 63.22 | 48.28 | 34.89 | 53.39 | 61.60 |
| DiffuDINO | 5 | 3 | 44.88 | 63.37 | 48.46 | 34.90 | 53.50 | 61.70 |
| DiffuDINO | 5 | 4 | 44.88 | 63.45 | 48.44 | 35.00 | 53.45 | 61.63 |
| DiffuDINO | 5 | 5 | 44.91 | 63.44 | 48.53 | 34.99 | 53.58 | 61.50 |
| DiffuDINO | 10 | 1 | 44.49 | 62.59 | 48.16 | 34.74 | 53.29 | 61.32 |
| DiffuDINO | 10 | 2 | 44.64 | 62.71 | 48.38 | 34.61 | 53.54 | 61.35 |
| DiffuDINO | 10 | 3 | 44.63 | 62.86 | 48.27 | 34.91 | 53.32 | 61.16 |
| DiffuDINO | 10 | 4 | 44.63 | 62.64 | 48.25 | 34.98 | 53.35 | 61.31 |
| DiffuDINO | 10 | 5 | 44.49 | 62.66 | 48.10 | 34.73 | 53.29 | 61.42 |

