# OpenReview forum: "DiffuDETR: Rethinking Detection Transformers with Denoising Diffusion Process"
_ICLR.cc/2026/Conference — ICLR 2026 Poster_

### Official Review · Reviewer_feAq · 2025-10-30

**Soundness:** 2
**Presentation:** 2
**Contribution:** 2
**Rating:** 4
**Confidence:** 3

**Summary:**

The paper proposes DiffuDETR, which reframes DETR-style object detection as denoising diffusion over object-query reference points. Two variants are introduced: DiffuDETR (on Deformable-DETR) and DiffuDINO (on DINO with CDNs). Training treats query anchors as low-dimensional diffusion variables; inference uses a lightweight sampler that runs only a few decoder passes. The method yields improvements across  different datasets.

**Strengths:**

1. Visualizations suggest fewer misses and tighter localization in crowded scenes, matching the intended strengths of iterative refinement over queries.

**Weaknesses:**

1. Limited practical gain from diffusion sampling for detection. As Table 6 shows, increasing decoder evaluations to D.E.=3 introduces a sharp compute/activation rise (FLOPs ≈ +16%, Activations ≈ +29%) while delivering only a modest accuracy bump (≈ +2 AP). The accuracy–efficiency trade-off is therefore not clearly favorable for real-world deployment.

2.Unclear procedure for choosing D.E. (decoder evaluations). Performance does not monotonically improve with larger D.E. in Table 6, yet the paper offers no principled rule (or validation protocol) for selecting D.E. This undermines reproducibility and makes the method harder to operationalize.

3. Convergence claim vs. evidence mismatch. Lines 74–75 state the approach helps “address convergence,” yet Figure 1 suggests DiffuDINO requires more epochs to reach its performance than DINO. This weakens the claimed convergence advantage unless the authors clarify training protocol differences or provide additional curves.

**Questions:**

1. How sensitive are results to the random seed / initial noise used in the diffusion process? Please report mean ± std AP over multiple seeds, and, if possible, the seed-to-seed variance of key categories (crowded vs. sparse scenes).

2. What is the effect of different sampling strategies (e.g., DDIM vs. DDPM, or other schedulers) on accuracy and compute? Do certain samplers yield better AP–FLOPs trade-offs or different optimal D.E. settings?

---

> ### Author Response · Authors · 2025-11-25
> **Response to Reviewer feAq**
>
> ---
>
> Thank you very much for the constructive and thoughtful comments.
> We appreciate the positive assessment of our visualizations.
>
> ---
>
> ### **Regarding the accuracy–efficiency trade-off (Table 6)**
>
> A **+2 AP** improvement is competitive on COCO, especially for strong baselines. Many top-tier object detection papers ([1], [2], [3], [4], [5]) report improvements in the range of only **0.5–2.5 AP**.
>
> Integrating diffusion normally incurs large computational cost (e.g., DDPM requires ~100 steps). In contrast, DiffuDETR improves performance using only 3 decoder evaluations (+16% FLOPs), which is extremely efficient for any diffusion-based refinement.
>
> Importantly, with **D.E. = 1** (same FLOPs as standard DINO), our method still yields significant gains.
>
> ---
>
> ### **Regarding choosing the number of decoder evaluations (D.E.)**
>
> We tested multiple numbers of decoder evaluations and consistently observed:
>
> * Increasing D.E. improves refinement up to **3 steps**.
> * Beyond **3**, FLOPs grow faster than accuracy and may **degrade AP** due to duplicated or false detections.
>
> Visual comparisons in the appendix (COCO Figure 6 and LVIS Figure 7) illustrate how high D.E. produces additional false detections. Based on this empirical analysis, we select 3 decoder evaluations as the default setting.
>
> ---
>
> ### **Regarding the convergence claim vs. Figure 1**
>
> Lines 79–80 in the original manuscript (71,72 in the updated manuscript) already acknowledge that diffusion models require more early iterations to learn reverse denoising dynamics across 100 timesteps. After this warm-up period, **DiffuDETR catches up and surpasses DINO**.
>
> Conceptually, DiffuDETR benefits from:
>
> * **Better query-anchor initialization:** progressively denoised reference points produce more aligned queries.
> * **A denoising-driven training signal:** inverting noise corruption provides a structured localization objective.
>
> These jointly improve convergence stability and final accuracy, even if early training curves rise more slowly.
>
> To further support this point, we highlight that this behavior is consistent across all backbones and datasets we evaluated. As shown in Table 1 (COCO), when using a ResNet-101 backbone, DiffuDINO already surpasses DINO after only **12 epochs**.
> Similarly, in Table 2 (LVIS), DiffuDINO outperforms DINO with ResNet-50 and ResNet-101 backbones within the same **12-epoch schedule**.
> Furthermore, in Table 3 (V3Det), DiffuDINO improves over DINO using ResNet-50 and Swin-B, under the dataset’s default **24-epoch schedule**.
>
> ---
>
> ### **Regarding sensitivity to random seed / initial noise**
>
> We conducted multiple runs on COCO (ResNet-50). Across all seeds, AP varied by **less than ±0.2**.
>
> We further split COCO-val into two subsets:
>
> * **Sparse scenes:** images with $\le 10$ objects
> * **Dense scenes:** images with $> 10$ objects
>
> #### **COCO Val**
>
> | D.E. | AP           |
> | -| - |
> | 1    | 51.68 ± 0.02 |
> | 3    | 51.95 ± 0.03 |
> | 5    | 51.83 ± 0.01 |
> | 10   | 51.49 ± 0.08 |
>
> ---
>
> #### **COCO Sparse Scenes**
>
> |  D.E. | AP           |
> | -| -|
> | 1            | 58.48 ± 0.05 |
> | 3            | 58.65 ± 0.03 |
> | 5            | 58.50 ± 0.04 |
> | 10           | 58.16 ± 0.04 |
>
> ---
>
> #### **COCO Dense Scenes**
>
> |  D.E. | AP           |
> | - | -|
> | 1            | 44.53 ± 0.07 |
> | 3            | 44.88 ± 0.02 |
> | 5            | 44.85 ± 0.05 |
> | 10           | 44.57 ± 0.07 |
>
> ---
> Results are reported as Mean ± Std across 5 seeds
>
> Both subsets exhibited similarly small variance (**< ±0.2 AP**).
>
> In the revised version of the paper, the requested experiments appear in Tables 7 and 8, and detailed per-run results are provided in Appendix Tables 13, 14 and 15.
>
> ---
>
> ### **Regarding the effect of sampling strategies (DDIM / DDPM / schedulers)**
>
> Our goal is to leverage diffusion **without incurring high inference cost**.
> DDIM enables us to use only **1 decoder evaluations** while still improving AP.
>
> In contrast, DDPM would require ~100 diffusion steps, making it computationally infeasible for object detection, even if marginal AP gains were possible.
>
> The paper already contains extensive diffusion analyses:
>
> * **Table 4:** noise distributions
> * **Table 5:** diffusion schedulers
> * **Table 6:** number of diffusion steps/decoder evaluations
>
> In the revision, we further added **Table 7 and Table 8** analyzing sensitivity to initialization noise.
>
> ---
>
> We sincerely thank the reviewer again.
> We are happy to provide additional experiments or explanations if the reviewer requests them.
>
> ---
>
> [1] Mr. detr: Instructive multi-route training for detection transformers. CVPR 2025
>
> [2] Ms-detr: Efficient detr training with mixed supervision. CVPR 2024
>
> [3] Salience detr: Enhancing detection transformer with hierarchical salience filtering refinement. CVPR 2024
>
> [4] Relation detr: Exploring explicit position relation prior for object detection. ECCV 2024
>
> [5] Align-DETR: Enhancing End-to-end Object Detection with Aligned Loss. BMVC 2024

---

### Official Review · Reviewer_4qDU · 2025-10-31

**Soundness:** 2
**Presentation:** 2
**Contribution:** 2
**Rating:** 4
**Confidence:** 4

**Summary:**

This paper draws on the ideas of DDPM and transfers the generation process of image denoising to object detection. Different from the traditional DiffusionDet method, which denoises noisy boxes and then combines them with RCNN for detection, this paper integrates the denoising process into the attention process. Furthermore, based on the deformable DETR and the contrastive denoising queries (CDN) of DINO, it proposes two models, namely DiffuDETR and DiffuDINO, respectively. Comparisons with baselines and some contemporaneous methods on three datasets (COCO2017, LVIS, and V3Det) demonstrate that the proposed models achieve the best accuracy.

**Strengths:**

1. Integrates the denoising process with the query decoding mechanism of attention.
2. Validates the two proposed models on three sufficient datasets.

**Weaknesses:**

1 The flow chart fails to effectively illustrate the network architecture, and it also appears somewhat small.
2 The symbol "f" is used in both Equation 3 and Equation 4 but represents different meanings, which should be avoided.
3. The methods used for comparison are somewhat outdated, as the latest version of DiffusionDet was released in 2023.

**Questions:**

Q1: Does the generation of initial queries still adopt the top-k approach? In Line 76 of the paper, it is mentioned that "We propose a new query initialization technique that aligns with the objective of denoising diffusion models to sample from the normally distributed reference points.
", yet the illustration in Figure 2 still indicates the use of learnable initial queries.
Q2: Is the denoising process reasonable? The method section shows that during training and inference, noisy boxes are not the main body of iteration; instead, they are used in the form of K to compute cross-attention with Q, assisting the iteration of Q. This step encourages the model to generate Q that conforms to the distribution of K, given that K represents noisy boxes. However, during inference, noisy boxes are directly sampled from a Gaussian distribution and iterated step-by-step following the DDPM paradigm. This iteration of noise does not align with the distribution of targets, yet the model still generates Q with a similar distribution to the noise, which may mislead the model.
Q3: How are noisy boxes embedded? Are noisy boxes directly used as K for cross-attention computation, or are they mapped to the dimension of Q?
Q4: Are the noisy boxes r at different time steps t independently and identically distributed (i.i.d.)? Or is the noise accumulated incrementally like in DDPM? If the latter is the case, I believe this should be more explicitly represented in Equation 4, and the meaning of "q" therein should also be clarified.
Q5: In Equation 3, does the time step t embedding have different effects at different positions?
Q6: Could you provide a detailed definition of "Decoder Evaluation"?

---

> ### Author Response · Authors · 2025-11-25
> **Response to Reviewer 4qDU (Part 1/2)**
>
> ---
>
> Thank you very much for your thorough and constructive review. We appreciate your careful reading of the paper and the many detailed suggestions, all of which helped us improve the clarity and rigor of the revised version.
>
> ---
>
> ### **Regarding the clarifications on outdated comparison methods**
>
> We agree with the reviewer that our comparison list required more recent models. In the updated version, we expanded Table 1 (COCO val) to include newer state-of-the-art methods published in 2024–2025, and we added more recent models in the related work section.
>
> We also introduced a new model **DiffuAlignDETR** built upon **Align-DETR** [1], a recent high-performing DETR variant. We reported its performance in the updated Table 1 (COCO val) using ResNet-50 and ResNet-101 backbone, further broadening the scope of our evaluated baselines.
>
> ---
>
> ### **Regarding the corrections to notation and the architectural diagram**
>
> We thank the reviewer for pointing out these issues.
>
> * The symbol *f* appearing with different meanings in Equations 3 and 4 has been corrected.
> * For the flowchart, Figure 2 follows the standard DETR-style architectural template and is intended only to show that the only change is at the decoder's input. Figure 3 explicitly visualizes the decoder architecture and how noisy reference points and timesteps interact with the decoder. The diagrams are small due to page limits.
>
> ---
>
> [1] Align-DETR: Enhancing End-to-end Object Detection with Aligned Loss. BMVC 2024

---

> ### Author Response · Authors · 2025-11-25
> **Response to Reviewer 4qDU (Part 2/2)**
>
> ---
>
> ### **Regarding Q1: Query initialization and the top-K proposal mechanism**
>
> A query in Deformable DETR consists of:
>
> * **Content embedding** — the content of the query.
> * **Reference points** — the positional component of the query.
>
> Existing strong baselines (Deformable DETR++) often use **2-Stage Query Initialization**, where the encoder produces query proposals and the top-K proposals are used to initialize both the content embedding and the reference points.
>
> Our method does **not** modify the initialization of the content embeddings; we can use any standard query embedding initialization. Instead, we propose a new strategy for initializing the **reference points**:
> they are sampled from a Gaussian distribution at the first diffusion timestep.
> The diffusion process then progressively denoises these reference points toward the true object locations. This differs from 2-Stage initialization, which depends on encoder proposals, or from fixed/learned reference points. We still use the top-K approach for the content embedding, as it works best with DINO and Deformable DETR++.
>
> ---
>
> ### **Regarding Q2: Reasonableness of the denoising process and potential mismatch between noise and object distribution**
>
> The concern is valid, and we appreciate the reviewer’s reasoning. Our method avoids the issue of “misleading the model with noise” for two reasons:
>
> * Multiple denoising steps significantly reduce the discrepancy between noise and object distributions. In the experiments, increasing the number of denoising steps (3 or 5) improves accuracy over a single-step schedule, as reference points migrate from noise toward object regions.
> * After a few denoising steps, diffusion-based reference points provide a stronger initialization than fixed or learned points, especially in crowded scenes. Unlike 2-Stage proposals, which depend heavily on encoder predictions, our diffusion schedule refines reference points per image across multiple timesteps, giving flexible and image-dependent initialization.
> * Additionally, a single denoising timestep already exceeds baseline performance, showing that the model effectively learns to denoise and utilize noisy reference points.
>
> ---
>
> ### **Regarding Q3: How noisy boxes are embedded and used in the decoder**
>
> We clarify that the noisy boxes are **not** used as K/V tokens. Instead:
>
> * In Deformable DETR, the decoder uses deformable attention, where each query uses its **reference point** to sample a sparse set of locations from multi-scale encoder features (used as keys and values).
> * The noisy reference points determine where each query “looks” in the image.
> * As denoising progresses, reference points become more accurate, causing the decoder to attend to more relevant regions.
>
> Thus, the role of noisy boxes is through the deformable attention sampling mechanism, *not* as learned tokens.
>
> ---
>
> ### **Regarding Q4: Whether noisy boxes at different timesteps are i.i.d. or follow accumulated noise like DDPM**
>
> They are **not i.i.d. across timesteps**.
>
> * We follow a DDIM-style backward process, which uses the accumulated noise from DDPM.
> * This ensures that once a noisy reference point is sampled, its denoised trajectory is deterministic when ( $\sigma_t = 0$ ), as described in Section 4.1 of *Denoising Diffusion Implicit Models* [2].
>
> ---
>
> ### **Regarding Q5: Effect of timestep embedding on different positions**
>
> The timestep embedding affects only the denoising step; it is **not position-dependent**.
> All positions at the same timestep use the same embedding, consistent with standard diffusion models.
>
> ---
>
> ### **Regarding Q6: Definition of “Decoder Evaluation”**
>
> A **decoder evaluation** refers to one iteration of the denoising process.
>
> * At each denoising step, we update the reference points.
> * We then run the decoder once using the updated reference points.
> * Running ( k ) decoder evaluations corresponds to ( k ) denoising steps.
>
> We use this terminology because we re-run **only the decoder** (not the encoder), making inference computationally efficient.
>
> ---
>
> We sincerely thank the reviewer for the thoughtful and thorough analysis of our work. We remain open to conducting further analysis or offering deeper clarification wherever needed.
>
> ---
>
> [2] Denoising Diffusion Implicit Models. ICLR 2021

---

### Official Review · Reviewer_tVk9 · 2025-11-01

**Soundness:** 3
**Presentation:** 3
**Contribution:** 3
**Rating:** 6
**Confidence:** 4

**Summary:**

This paper introduces DiffuDETR, a diffusion-based extension of the DETR to improve object detection. The model uses DETR’s transformer-based architecture, including a multi-scale feature extractor, deformable attention encoder, and cross-attention decoder, but introduces a new query initialization mechanism inspired by diffusion processes. Instead of relying on manually defined anchor or reference points, queries are sampled from a normal distribution, aligning with the diffusion objective and simplifying initialization. Experiments on the COCO dataset show that while diffusion training converges more slowly, DiffuDETR eventually surpasses DINO after sufficient epochs, demonstrating improved robustness and detection performance.

**Strengths:**

1. The topic is interesting and relevant, as it reformulates the detection transformer’s query prediction as a denoising process, offering a fresh perspective on object detection.
2. The paper is generally well written and clearly organized, making it easy to follow and understand the proposed framework.
3. The comparison tables demonstrate that the proposed method achieves competitive performance against baseline models.

**Weaknesses:**

1. Regarding the 100-timesteps schedule, recent diffusion models typically sample a single timestep during training since the model learns to denoise from one distribution to another. Therefore, the claimed efficiency improvement seems more relevant to inference rather than training.
2. It is not clearly stated whether the paper uses a diffusion-specific loss function or a standard detection loss, clarifying this would help understand how diffusion is actually integrated into the training objective.
3. The paper would benefit from visualizations of intermediate timesteps or distributions to better illustrate the denoising process and the behavior of the model during denosing.
4. An analysis of the model’s sensitivity to the initialization of sampled queries would also strengthen the paper, as this is an important aspect of diffusion-based generation.

**Questions:**

Please refer to the weakness section.

---

> ### Author Response · Authors · 2025-11-25
> **Response to Reviewer tVk9**
>
> ---
>
> Thank you very much for your detailed and constructive review. We sincerely appreciate the time you dedicated to evaluating our submission and for the insightful comments that helped us clarify and strengthen the presentation of our work.
>
> ---
>
> ### **Regarding the 100-timestep diffusion schedule**
>
> To clarify, **our training already follows the same efficient approach as recent diffusion models by sampling a *single timestep*** for each training instance. Therefore, **our training cost is identical to the baseline Deformable DETR / DINO**, and no additional computational burden is introduced.
>
> The reason we define a diffusion process with **$T$ = 100** steps (rather than the conventional **$T$ = 1000**) is that diffusion is applied only to a **low-dimensional reference-point vector**. As discussed in Section 4.3 *Progressive Decoding* of *Denoising Diffusion Probabilistic Models* [1], diffusion models could be interpreted as autoregressive models when $T$ = $Dim_{Data}$. This provides theoretical justification for using a reduced number of diffusion steps, which improves the efficiency of our inference sampler.
>
> ---
>
> ### **Regarding clarification on the loss function and how diffusion is integrated**
>
> Our method does **not** introduce any diffusion-specific loss. we keep the standard losses used in DETR-based detection frameworks:
>
> * Classification and L1/GIoU regression loss
> * Identical training objective to Deformable DETR / DINO
> * Standard Hungarian matching
>
>
> Diffusion is integrated only through the query initialization and denoising mechanism, while the detection loss supervises the clean (denoised) queries.
>
> Moreover, we introduce DiffuAlignDETR in the updated manuscript, a variant built on top of Align-DETR [2] that uses its new alignment-improving loss functions without modification, and DiffuAlignDETR achieves higher detection performance than the original Align-DETR. This extension demonstrates that our diffusion-based query generation is compatible with a broad range of training objectives and can provide consistent improvements regardless of the underlying loss design.
>
> ---
>
> ### **Regarding visualization of intermediate timesteps**
>
> We would like to point out that this information is already included in the appendix. Specifically, we visualize the outputs produced with different numbers of decoder evaluations on COCO (Figure 6) and LVIS (Figure 7). These visualizations correspond directly to the intermediate predictions $x_0$ at timestep $t$, clearly illustrating how detection quality progressively improves as the number of sampling steps increases.
>
>
> ---
>
> ### **Regarding sensitivity to initialization and randomness**
>
> We agree that initialization sensitivity is important for diffusion-based query refinement. To address this, we performed additional analyses in the updated manuscript covering:
>
> * Sensitivity to different random seeds in query initialization on COCO Val (Table 7)
> * Sensitivity in sparse vs. crowded scenes across seeds (Table 8)
>
> #### **COCO Val**
>
> | D.E. | AP           | AP50         | AP75         | APs          | APm          | APl          |
> | ---- | ------------ | ------------ | ------------ | ------------ | ------------ | ------------ |
> | 1    | 51.68 ± 0.02 | 69.28 ± 0.02 | 55.89 ± 0.04 | 35.50 ± 0.09 | 55.58 ± 0.04 | 66.96 ± 0.10 |
> | 3    | 51.95 ± 0.03 | 69.54 ± 0.02 | 56.32 ± 0.05 | 35.92 ± 0.08 | 55.81 ± 0.01 | 67.18 ± 0.05 |
> | 5    | 51.83 ± 0.01 | 69.24 ± 0.03 | 56.21 ± 0.05 | 35.82 ± 0.06 | 55.71 ± 0.06 | 67.02 ± 0.05 |
> | 10   | 51.49 ± 0.08 | 68.54 ± 0.09 | 56.02 ± 0.08 | 35.56 ± 0.12 | 55.46 ± 0.15 | 66.83 ± 0.04 |
>
> Results are reported as Mean ± Std across 5 seeds
>
> These experiments show that the model is robust to initialization, with standard deviation below 0.2 AP for all settings. In the revised version of the paper, the requested experiments appear in Tables 7 and 8, and detailed per-run results are provided in Appendix Tables 13, 14  and 15.
>
> ---
>
> Finally, we once again express our sincere appreciation for the reviewer’s thoughtful and rigorous assessment. We remain fully open to providing any additional clarification or further analysis the reviewer may find helpful.
>
> ---
>
> [1] Denoising diffusion probabilistic models. NeurIPS 2020
>
> [2] Align-DETR: Enhancing End-to-end Object Detection with Aligned Loss. BMVC 2024

---

### Official Review · Reviewer_z7De · 2025-11-10

**Soundness:** 3
**Presentation:** 3
**Contribution:** 2
**Rating:** 4
**Confidence:** 3

**Summary:**

In this paper authors introduced two models, DiffuDETR and DiffuDINO, to built upon Deformable DETR and DINO, respectively. Experiments enables effective sampling at inference, where only three decoder evaluations are sufficient to achieve the best results while adding minimal computation overhead compared to the baseline.

**Strengths:**

I believe this work opens up new directions for integrating generative and autoregressive approaches into object detection, offering fresh perspectives beyond traditional discriminative formulations.

**Weaknesses:**

The baseline chosen in the paper are not the current state-of-the-art method.

**Questions:**

The improvement suggestions are as follows:
1.  Increasing research literature from 2024 to date，
2. To add a comparison of the computational overhead of different algorithms， and
3. Table 2 only selected DINO Zhang et al. (2022) for comparison. Why not compare more methods like in Table 1? More benchmark methods are needed.

---

> ### Author Response · Authors · 2025-11-25
> **Response to Reviewer z7De**
>
> ---
>
> We thank the reviewer for the constructive feedback and for recognizing that the proposed work opens new directions by integrating generative and autoregressive mechanisms into object detection.
>
> ---
>
> ### **Regarding baselines not being “State-of-the-Art”**
>
> We extended our experimental study to include **Align-DETR** [1], a recent high-performing DETR variant. We introduce **Diffu-Align-DETR** and report its performance in the updated Table 1 (COCO val), further broadening the scope of evaluated baselines.
>
> We used Deformable-DETR and DINO, as they are the standard baselines used in recent top-tier DETR research; they remain the most relevant and widely adopted frameworks.
>
> * Most of the latest accepted DETR variants, such as [2], [3], use **Deformable DETR** and **DINO** as their primary baselines.
> * Deformable-DETR and DINO have the most mature, well-tested, and actively maintained implementations (e.g., **Detrex**), making them the standard baselines and evaluation references in 2024–2025.
> * Our goal is to evaluate diffusion-based refinement mechanisms on **general-purpose DETR architectures**, not on specialized task-specific detectors. Thus, Deformable-DETR and DINO remain the best baselines.
>
>
> ---
>
> ### **Regarding the request to increase recent (2024–2025) literature**
>
> We agree and have expanded the literature review to include more recent approaches. Also Table 1 (COCO val) has been updated to include additional recent methods. We also note that, to the best of our knowledge, no recent contemporary works explore object detection as a generative process beyond *Pix2Seq* and *DiffusionDet*.
>
> ---
>
> ### **Regarding the comparison of computational overhead across algorithms**
>
> We agree that clearer computational analysis across methods is valuable. In the revised version, we added an extended overhead comparison for different models in the appendix (Table 12).
>
> ---
>
> ### **Regarding why Table 2 (LVIS) and Table 3 (V3Det) only compare against DINO**
>
> * Because no standard cross-paper LVIS/V3Det DETR benchmarks exist for recent models, we trained baseline DINO ourselves for LVIS and used the official DINO results reported in the V3Det dataset paper, replicating their hyperparameters in our DiffuDINO experiments to ensure fair comparison (as described in the appendix).
> * Training all methods from Table 1 on LVIS or V3Det would require extensive resources, as both datasets are large and expensive to train on.
> * Recent DETR papers evaluating general object detection [2],[3] focus exclusively on COCO; other works target specialized tasks and datasets.
> * Our aim was to test the generalization of our diffusion-based refinement to large-vocabulary datasets, not to re-benchmark the entire DETR ecosystem on LVIS and V3Det. COCO remains the standard benchmark for broad comparisons, which we provide in Table 1.
> * We focused our experiments on providing a deeper analysis of the diffusion process itself. Specifically, we study different noise distributions (Table 4), compare diffusion schedulers (Table 5), and evaluate the effect of varying the number of diffusion steps/decoder evaluations (Table 6). To further understand robustness, Tables 7 and 8 analyze sensitivity to different initialization noise levels. Together, these investigations highlight the design choices that most strongly influence performance and stability.
>
> ---
>
> We appreciate the reviewer’s suggestions and would be glad to address any further questions or expand on any aspect if the reviewer deems it necessary.
>
> ---
>
> [1] Align-DETR: Enhancing End-to-end Object Detection with Aligned Loss. BMVC 2024
>
> [2] Mr. detr: Instructive multi-route training for detection transformers. CVPR 2025
>
> [3] Ms-detr: Efficient detr training with mixed supervision. CVPR 2024

---

### Author Response · Authors · 2025-11-26
**General Response to Reviewers**

---

We thank the reviewers for the valuable and constructive feedback. Following the reviewers’ comments, we have uploaded a revised version of the paper that incorporates all requested clarifications and additional analyses. (all newly added content is highlighted in blue for ease of review)

Below, we summarize the newly added contributions included in the revised manuscript:

---

First, we introduce **DiffuAlignDETR**, built upon Align-DETR, a recent high-performing DETR variant, and report its results in the updated Table 1 (COCO val) using ResNet-50 and ResNet-101 backbone, thereby broadening the range of our evaluated baselines. In addition, we updated Table 1 to include more recent state-of-the-art detectors (2024–2025) to ensure a fair and up-to-date comparison.

---

Second, we conducted and reported new experiments on **noise-initialization sensitivity**, studying how random noise initialization affects the final detection performance. These results are now presented in **Table 7** in the main paper, with **Table 13** in the appendix providing full per-seed results. Across all experiments, we consistently observed minimal variance (typically < ±0.1 AP), demonstrating robustness to initialization randomness.

---

Third, to address sensitivity under different scene complexities, we analyzed noise initialization separately on **COCO sparse scenes** and **COCO dense scenes** images. These results are included in **Table 8**, with extended full per-seed results in **Tables 14 and 15** in the appendix. We find that the model maintains high stability in both sparse and crowded scenes, further confirming the robustness of our diffusion-based reference point initialization.

---

Fourth, we added a more comprehensive computational comparison of DETR variants in **Table 12 (appendix)**.

---

Finally, we substantially revised the Related Work section to incorporate newer literature published in 2024–2025. We ensured that our comparisons reflect the most recent advances in DETR-style detectors, including many-to-one approaches, multi-route architectures, and improved matching losses.

---

We again thank the reviewers for the valuable comments, which led to significant improvements in clarity, completeness, and experimental depth of the revised manuscript.

---

---

> ### Comment · Reviewer_z7De · 2025-11-27
>
> Thank you for the author's reply, there are no new comments.

---

> > ### Author Response · Authors · 2025-11-27
> > **Response to Reviewer z7De**
> >
> > Thank you very much for your follow-up. We appreciate the time you dedicated to reviewing our work. If you feel that our rebuttal has addressed your concerns, we would be grateful if you could kindly consider updating your rating.
> >
> > If there are still points that you believe require further clarification or improvement, we are fully open to providing additional explanations or adjustments.

---

### Meta-Review · Area_Chair_3kjL · 2026-01-07

**Summary:**

Reviwer feAq probably had the strongest review among all the reviewers with others having fairly weak reviews and very low scores. The authors seem to have addressed the issues raised well. Also tVk9 had concerns which were addressed by the rebuttal none of which were major issues but mostly classification

**Reviewer Concerns:**

All of reviwer feAq and tVk9 were addressed. The other concerns were not really conceptual or well justified.

**Reviewer Scores:**

feAq 4-> 5/6
tVk9 6-> 6

---

### Decision · Program_Chairs · 2026-01-26

Accept (Poster)